# Healthcare provider's adherence to immediate postpartum care guidelines in Gondar province hospitals, northwest Ethiopia: A multicenter study

**Azmeraw Ambachew Kebede** [1]*, **Birhan Tsegaw Taye** [2], **Kindu Yinges Wondie** [1], **Agumas Eskezia Tiguh**[1], **Getachew Azeze Eriku**[3], **Muhabaw Shumye Mihret** [1]

1 Department of Clinical Midwifery, School of Midwifery, College of Medicine and Health Sciences, University of Gondar, Gondar, Ethiopia, 2 Department of Clinical Midwifery, College of Medicine and Health Sciences, Debre Berhan University, Debre Berhan, Ethiopia, 3 Department of Physiotherapy, College of Medicine and Health Sciences, University of Gondar, Gondar, Ethiopia

* azmuzwagholic@gmail.com

**Data Availability Statement:** All relevant data are within the manuscript and its Supporting information files.

## Abstract

### Background

The immediate postpartum period is the most critical time for both the mother and the newborn. However, it is the most neglected part of the maternal continuum of care, and evidence in this regard was scarce in Ethiopia. Therefore, this study aimed to assess the healthcare provider's adherence to immediate postpartum care guidelines and associated factors in hospitals of Gondar province.

### Methods

A multicenter observational cross-sectional study was conducted among 406 healthcare providers from 15th November 2020 to 10th March 2021. Data were collected through face-to-face interviews and direct observation using a structured questionnaire and standardized checklist respectively. Data was entered into EPI INFO 7.1.2 and analyzed by SPSS version 25. Both bivariable and multivariable logistic regression analyses were carried out. The level of significance was declared based on the adjusted odds ratio (AOR) with a 95% confidence interval (CI) at a p-value of $\leq$ 0.05.

### Results

Overall, 42.4% (95% CI: 37.5, 47.2) of healthcare providers had complete adherence to immediate postpartum care guidelines. Having birth assistant (AOR = 1.87; 95% CI: 1.10, 9.67), being married (AOR = 1.59; 95% CI: 1.15, 3.31), availability of postpartum care guidelines at the maternity ward (AOR = 2.39; 95% CI: 1.44, 3.98), received basic emergency obstetric and newborn care (BEmONC) training (AOR = 2.1; 95% CI: 1.2, 3.6), monthly income of $\geq$ 10001 Ethiopian birr (AOR = 3.55; 95% CI: 1.30, 9.67), and work experience

**Funding:** The authors received no specific funding for this work.

**Competing interests:** The authors have declared that no competing interests exist.

**Abbreviations:** AOR, adjusted odds ratio; BEmONC, basic emergency obstetric and newborn care; CI, confidence interval; COR, crude odds ratio; ETB, Ethiopian birr; IPPCG, immediate postpartum care guidelines; IPPP, immediate postpartum period; PNC, postnatal care; SPSS, statistical package for social science; SSA, Sub-Saharan Africa; VIF, variance inflation factor.

of $\geq$ 6 years (AOR = 0.15; 95% CI: 0.06, 0.38) were significantly associated with healthcare providers adherence to immediate postpartum care guidelines.

## Conclusion

This study indicated that health worker's adherence to immediate postpartum care guidelines was low. Hiring adequate health workers, availing postpartum guidelines at the maternity ward, improving the salary and education opportunities for healthcare workers of healthcare workers, and provision of BEmONC training will have a great role in improving healthcare provider's adherence to immediate postpartum care guidelines.

## Introduction

The immediate postpartum period (IPPP) is the first 24 hours after delivery [1]. The IPPP is very dangerous for both the mother and the newborn [2]. This time in particular will determine the long-term health and well-being of neonates [3, 4]. However, this IPPP is neglected by many health professionals and is rarely given attention [1].

In 2017, the global maternal mortality was about 295,000 (an estimated 211 maternal deaths per 100,000 live births) [5] and 2.5 million neonatal deaths [6]. Of these, 94 percent of all maternal deaths and 80% of neonatal deaths occur in developing countries [6, 7]. From the global deaths, Sub-Saharan Africa (SSA) accounted for more than two-thirds of the estimated maternal deaths (i.e., 542 maternal deaths per 100, 000 live births) [8]. About 20 to 44% of these maternal deaths occur during the postpartum period (PPP) [8]. This can be reversed by providing appropriate and timely postnatal care (PNC) by qualified health professionals [4]. This is because delays in getting quality immediate PNC services are one of the main reasons for maternal death [9].

Sub-Saharan Africa faces many challenges in ensuring the optimal health of both the mother and newborn [10]. One of the reasons is inequitable and discriminatory maternal health service provision based on the specific status of women [11]. Ethiopia is one of the countries in SSA with the highest maternal and neonatal mortality rate and a low rate of access to maternal health services. Thus, as stated in the 2016 Demographic Health Survey of Ethiopia (EDHS 2016), 32% of women use antenatal care (ANC) four or more times, 26% of women gave birth in health institutions, and 17% of women received PNC [12]. Besides, according to the 2017 Fragile States Index, Ethiopia is one of the 15 countries grouped under the "very high alert" category [5].

Although the maternal and neonatal mortality rates in Ethiopia have been declining since 2000, the reduction is still unsatisfactory. Thus, the Maternal Mortality Ratio (MMR) was 871 in 2000 [13], 673 in 2005 [14], 676 in 2011 [15], and 412 per 100,000 live births in 2016 [12]. Besides, neonatal mortality rate (NMR) has dropped from 49 in 2000 to 39 in 2005 [13, 14], and 37 in 2011 to 29 per 1000 live births in 2016 [12, 15]. In Ethiopia, as a country, a lot of works has been done to increase maternal healthcare service utilization with the promise of reducing the maternal mortality ratio from 412 per 100,000 live births to 199, and the neonatal mortality rate from 29 to 10 per 1000 live births by 2020 [16]. However, it is impossible to achieve this goal without the healthcare provider's adherence to maternal and newborn guidelines.

Moreover, the failure of healthcare providers to comply with immediate postpartum care guidelines (IPPCG) and postpartum counseling contributes to poor PNC uptake. Evidence

supports that the right steps taken by health professionals can greatly reduce maternal deaths, principally during the IPPP [17]. To reduce maternal and neonatal mortalities on a permanent and sustainable basis, healthcare providers working in the maternity unit play the lion's share [18]. Midwives in particular are the frontline providers of healthcare for women and newborns [19]. Empirical evidence showcases that an estimated 41% of maternal and 39% of neonatal deaths can be prevented by expanding midwifery-delivered interventions [18]. In addition, ensuring quality midwifery care is a prerequisite for achieving the sustainable development goals (SDG), in particular targets 3.1 and 3.2 [20, 21].

Studies examining the healthcare provider's adherence to IPPCG are scarce. Most of the previously published literature focuses on women's utilization of PNC and associated factors. Only one study has been conducted in Ethiopia regarding healthcare provider's adherence to IPPCG. According to this study, 22.8% of healthcare providers had complete adherence to IPPCG [22]. But, while the study intended to assess healthcare provider's adherence to IPPCG, the data were collected from the women. It fails to assess which specific factors will affect the adherence level of healthcare providers to the existing IPPCG. Therefore, this is one of the first studies to ascertain the healthcare provider's adherence to IPPCG as to the authors' deep review. Also, this study aimed to identify individual, workplace, and profession-related factors associated with healthcare provider's adherence to the IPPCG in Gondar province hospitals, northwest Ethiopia.

## Methods and materials

### Study design, area, and period

A multicenter institution-based cross-sectional study was conducted from 15[th] November 2020 to 10[th] March 2021. The study was conducted at Hospitals in Gondar province Amhara national regional state, northwest Ethiopia. In the Gondar province, there are four zones, namely South Gondar, Central Gondar, West Gondar, and North Gondar zones. There are a total of 22 hospitals in Gondar province. Among these, the University of Gondar and Debre Tabor are referral hospitals. The remaining 20 hospitals are primary hospitals except Debark general hospital. These hospitals are serving more than 10 million population in the four zones of Gondar province and surrounding zones like North Wollo and Waghimra zone.

### Study population

All healthcare providers working at the maternity ward and attending women at or after 28 weeks of gestation (i.e. after fetal viability in the Ethiopian context) were the study population.

### Sample size determination and sampling procedure

The sample size for this study was determined by using a single population proportion formula by considering the following assumptions: 95% level of confidence, 50% provider's adherence to immediate postpartum care guidelines, and 5% margin of error.

$$\text{n} = \frac{(Z\alpha/2)^2 p(1-p)}{d^2} = \text{n} = \frac{(1.96)^{2*}0.5\,(1-0.5)}{(0.05)2} = 384$$

Where, n = required sample size, $\alpha$ = level of significant, z = standard normal distribution curve value for 95% confidence level = 1.96, p = proportion of healthcare providers adherence to IPPCG, d = margin of error. After considering a non-response rate of 10%, we obtained a total sample size of 422. Data were collected from 15 hospitals (2 tertiary hospitals, 1 general hospital, and 12 primary hospitals). The seven primary hospitals were excluded due to their

very low monthly delivery report. The lists of healthcare providers were obtained from each hospital and the sampling frame was designed by numbering the list of healthcare providers. Then, the total sample size was distributed to each selected hospital proportionally. Finally, the participants were selected by a simple random sampling technique using a table of random generation.

## Variables of the study

The dependent variable for this study was adherence to IPPCG whereas the independent variables were age, sex, educational level, marital status, availability of smartphone or computer, and availability of media, year of experience, relation to the nearby boss, intention to stay in the profession, job satisfaction, facility type, availability of internet, availability of postnatal guideline in the ward, working time, training on essential newborn care, training on basic emergency obstetric and newborn care (BEmONC), presence of regular follow-up by the manager, workload or shortage of staff in the delivery room, and location of the health facility.

## Operational definitions and measurements

Immediate postpartum care: a total of 17 checklists or questions were prepared to assess healthcare providers adhere to the IPPCG. Each checklist has a "yes "or "no" response giving a score of 0–17 (i.e., a score of 1 was given for "yes" and 0 for "no" response). Likewise, the healthcare provider's adherence to IPPCG was dichotomized as complete adherence if the healthcare provider performs all the offered checklists (which was coded as "1") and incomplete adherence if the provider fails to perform all the offered checklists (which was coded as "0"). The immediate postpartum period is 6 hours after delivery in the Ethiopian context [22].

**Job satisfaction.** A total of 8 questions were prepared to assess the study participates job satisfaction level, accordingly, healthcare providers who scored above the mean score were considered as satisfied, otherwise dissatisfied [23].

## Data collection tools, methods, and procedures

The data collection tool was developed by reviewing literature and guidelines [22, 24] and data were collected using a structured questionnaire and checklists through face-to-face interviews and direct observation respectively. The questionnaire was assessed by a group of researchers to evaluate and enhance the items in the question. The questionnaire contains socio-demographic characteristics, professional and environment-related factors, and questions assessing the provider's adherence to IPPCG. About 20 personnel were recruited for the data collection process. These include 15 Diploma midwives for data collection and 5 BSc midwives for supervision. During the actual data collection, one healthcare provider was observed only once. In the meantime, we tried to minimize the observation bias (the Hawthorne effect) of healthcare provider's practice by telling the healthcare providers in which the data collection process is mysterious and can no longer be reported to their immediate supervisors or publicly shared. Healthcare providers were also informed of the objective of the study, which was designed to improve healthcare provider's compliance with the IPPCG, rather than make judgments about practices performed during the data collection process. Also, healthcare providers do not know what detailed procedures and activities are included in the questionnaire, so they cannot prepare in any way. Moreover, the data collectors were from other health facilities, not from the same health facility in order to minimize the effect of personal relationships.

## Data quality control

Before the actual data collection, a pretest was done on 5% of the calculated sample size (22 healthcare providers) outside of the study area. The data collectors and supervisors were trained about the interview technique and overall data collection process for 3 days. During data collection, the questionnaire was checked for completeness by the supervisors.

## Data processing and analysis

Data were checked and coded manually. EPI INFO version 7.1.2 and SPSS version 25 were used for data entry and analysis respectively. Descriptive statistics were used to present participants' characteristics and adherence to the IPPCG. The binary logistic regression model was fitted and bivariable and multivariable logistic regression analyses were undertaken. Variables having a p-value of less than 0.2 at a bivariable logistic regression analysis were candidates for multivariable logistic regression analysis. Multicollinearity was screened using the variance inflation factor (VIF), in which VIF less than 10 was acceptable. In the multivariable logistic regression, the level of significance was claimed based on AOR with 95% CI at a p-value of $\leq 0.05$.

## Ethical considerations

The study was conducted under the Ethiopian Health Research Ethics Guideline and the declaration of Helsinki. Ethical clearance was obtained from the Institutional Ethical Review Board (IRB) of the University of Gondar (**Reference number: V/P/RCS/05/413/2020**). A formal letter of administrative approval was gained from each selected hospital. Written informed consent was taken from each of the study participants after a clear explanation of the aim of the study.

# Results

## Socio-demographic characteristics

In this study, a total of 422 healthcare providers were recruited; but only 406 healthcare providers were included in the study, giving a response rate of 96.2%. The mean age of the study participants was 28.4 years (SD ±4.7) and 60.8% of the participants were between the age group of 26 to 30 years. The majority of the healthcare providers were Degree midwives with an average monthly income of 5861.3 Ethiopian Birr (ETB) [Table 1].

## Workplace and profession-related characteristics

Of the total study participants, more than three-fourths were satisfied with their profession and 71.4% of the healthcare providers had an intention to stay in their profession. About 31% and 38.4% of the participants received BEmONC and essential newborn care training. Eighty-three percent of the healthcare providers had a good interest to work in the delivery room and 77.3% of the participants had good relations with the nearby manager [Table 2].

## Healthcare providers' adherence to immediate postpartum care guidelines

The proportion of healthcare provides completely adhering to IPPCG was 42.4% (95% CI: 37.5, 47.2). Tetracycline eye ointment application (96.6%), oxytocin administration (95.6%), and measuring the weight of the newborns (95.1%) were areas of good practice. The level of adherence to IPPCG varied across individuals [Table 3].

**Table 1. Socio-demographic characteristics of study participants in Gondar province hospitals, northwest Ethiopia, 2020/2021 (n = 406).**

| Characteristics | Category | Frequency | Percentage (%) |
|---|---|---|---|
| **Age of participant in year** | ≤25 | 85 | 20.9 |
| | 26–30 | 247 | 60.8 |
| | ≥31 | 74 | 18.2 |
| **Sex of the participant** | Male | 272 | 67 |
| | Female | 134 | 33 |
| **Current marital status** | Single | 164 | 40.4 |
| | Married | 242 | 59.6 |
| **Work experience** | ≤2 | 140 | 34.5 |
| | 3–5 | 210 | 51.7 |
| | ≥6 | 56 | 13.8 |
| **Having smart phone or computer** | Yes | 256 | 63.1 |
| | No | 150 | 36.9 |
| **Ever watching television** | Yes | 353 | 86.9 |
| | No | 53 | 13.1 |
| **Ever reading newspaper** | Yes | 207 | 51 |
| | No | 199 | 49 |
| **Professional category** | Midwifery diploma | 119 | 29.3 |
| | Midwifery Bachelor degree | 243 | 59.8 |
| | Midwifery master's degree | 25 | 6.2 |
| | Others* | 19 | 4.7 |
| **Monthly income** | <5000 ETB | 140 | 34.4 |
| | 5001–10000 ETB | 239 | 58.9 |
| | >10001 ETB | 27 | 6.7 |

Note:

* GP, Residents, and IESO; ETB, Ethiopia Birr;

## Factors associated with adherence to immediate postpartum care guidelines

Multivariable logistic regression analysis revealed that marital status of healthcare providers, presence of postpartum care guidelines in the maternity ward, presence of an assistant for the delivery, received training on BEmONC, and average monthly income were factors significantly associated with healthcare provider's adherence to IPPCG.

Healthcare providers who got marriage were 1.59 times more likely to completely adhere to IPPCG as compared to their counterparts (AOR = 1.59; 95% CI: 1.15, 3.31). Likewise, healthcare providers who had an assistant for delivery were 1.87 times more likely to have had complete adherence to IPPCG as compared to those healthcare providers who hadn't have a birth assistant for the delivery (AOR = 1.87; 95% CI: 1.10, 9.67).

The odds of adhering to IPPCG were 2.39 times higher among healthcare workers who had postpartum care guidelines in their hospital than their counterparts (AOR = 2.39; 95% CI: 1.44, 3.98). Similarly, the health care providers who received BEmONC training were 2.1 times more likely to adhere to IPPCG as compared to the reference group (AOR = 2.1; 95% CI: 1.2, 3.6). Moreover, the likelihood of adhering to IPPCG was 3.55 times higher among healthcare providers earning ≥ 10001 ETB than those professionals with lesser monthly income (AOR = 3.55; 95% CI: 1.30, 9.67).

**Table 2. Workplace and professional related characteristics of study participants in Gondar province hospitals, northwest Ethiopia, 2020/2021 (n = 406).**

| Characteristics | Category | Frequency | Percentage (%) |
|---|---|---|---|
| **Facility type** | Primary hospital | 214 | 52.7 |
| | General hospital | 32 | 7.9 |
| | Tertiary hospital | 160 | 39.4 |
| **Facility location** | Urban | 223 | 54.9 |
| | Semi-urban | 183 | 45.1 |
| **Self-rated relation with the nearby manager** | Good | 314 | 77.3 |
| | Poor | 92 | 22.7 |
| **Satisfaction on the profession** | Satisfied | 324 | 79.8 |
| | Unsatisfied | 82 | 20.2 |
| **Availability of internet in the hospital** | Yes | 232 | 57.1 |
| | No | 174 | 42.9 |
| **Working time** | Day | 295 | 72.7 |
| | Night | 111 | 27.3 |
| **Assistant present for the delivery** | Yes | 318 | 78.3 |
| | No | 88 | 21.7 |
| **PNC guidelines in the maternity ward** | Yes | 218 | 53.7 |
| | No | 188 | 46.3 |
| **Received BEmONC training** | Yes | 126 | 31 |
| | No | 280 | 69 |
| **When did you take the training (n = 126)** | Within 2 years | 66 | 52.4 |
| | Before 2 years | 60 | 47.6 |
| **Intention to stay in the profession** | Yes | 290 | 71.4 |
| | No | 116 | 28.6 |
| **Interest to work in the delivery room** | Yes | 340 | 83.7 |
| | No | 66 | 16.3 |
| **Workload load in the ward** | Yes | 179 | 44.1 |
| | No | 227 | 55.9 |
| **Working part-time at private health facility** | Yes | 51 | 12.6 |
| | No | 355 | 87.4 |
| **Regular checkup by the nearby manager** | Yes | 287 | 70.7 |
| | No | 119 | 29.3 |
| **Education while working** | Yes | 151 | 37.2 |
| | No | 255 | 62.8 |
| **Training on essential newborn care** | Yes | 156 | 38.4 |
| | No | 250 | 61.6 |
| **When did you take the essential newborn care training** | Within 2 years | 102 | 65.4 |
| | Before 2 years | 54 | 34.6 |

Note: PNC, postnatal care; BEmONC, basic emergency obstetric and neonatal care

Lastly, healthcare providers with 6 years of work experience or more were 85% times less likely to be completely adhering to IPPG as compared with healthcare providers with work experience of 2 years or less (AOR = 0.15; 95% CI: 0.06, 0.38) [Table 4].

## Discussion

Healthcare providers are strongly encouraged to adhere to IPPCG, so that any complications on the neonates and mothers will be recognized and managed in a timely manner. These

**Table 3. Adherence to immediate postpartum care checklist among health care providers at hospitals in Gondar province northwest Ethiopia, 2020/2021.**

| Immediate postpartum checklists | Frequency |
|---|---|
| APGAR score assessed | 365 (89.9%) |
| Thermal protection ensured | 318 (78.3%) |
| Sex of neonate recorded | 360 (88.7%) |
| Oxytocin provided for the woman | 388 (95.6%) |
| Placental completeness checked | 317 (78.1%) |
| Uterine massage done | 352 (86.7%) |
| Early breastfeeding initiated | 363 (89.4%) |
| Counsel on breastfeeding attachment and positioning | 321 (79.1) |
| Vitamin K given for the neonate | 373 (91.9%) |
| Tetracycline eye ointment given | 390 (96.1%) |
| Head circumference measured | 199 (49%) |
| Length of the newborn measured | 198 (48.8%) |
| Weight of the newborn measured | 386 (95.1%) |
| Maternal vital signs checked | 318 (78.3%) |
| Counsel on danger signs and immunization | 309 (76.1%) |
| Proper documentation done | 341 (84%) |

timely problem identifications and interventions could then reduce maternal and neonatal mortality occurring during the IPPP on a sustainable basis. Hence, the current study aimed at assessing healthcare provider's adherence to IPPCG and associated factors in hospitals of Gondar province, northwest Ethiopia. Accordingly, the proportion of healthcare providers adhering to IPPCG was 42.4% and independently predicted by the availability of birth assistants, married marital status, availability of postpartum care guidelines at the maternity ward, provision of BEmONC training, and monthly income of $\geq$ 10001 ETB.

In the present study, about two-fifths (42.4%) of healthcare providers adhered to IPPCG. This finding is higher than a study conducted in Mekelle, Ethiopia-22.8% [22]. This higher proportion might be ascribed to maternal and neonatal health is a major public health issue and getting attention worldwide. Hence, training is being given regularly for healthcare providers regarding maternal and neonatal health so as to increasing their knowledge, skill, and self-efficacy [25]. In addition to increasing their knowledge and skill, training might be a source of income and used as entertainment and recreation for providers. The other most important contributing factor might be the Ethiopian Midwife Association (EMwA) is doing an incredible effort for midwives such as continuing professional development across the country. Ethiopian midwife association in collaboration with the Ethiopian ministry of health and other non-governmental organization are doing many activities including giving training for healthcare providers and doing researches that further enhance the growing up of the midwifery profession [26, 27].

The adherence level of healthcare workers towards tetracycline eye ointment application, oxytocin provision, and weight measurement was 96.6%, 95.6%, and 95.1% respectively. The result of this study is higher as compared to the findings from Mekelle, in which healthcare provider's adherence to tetracycline application-67.7%, oxytocin administration-85.1%, and weight measurement-91.1% [22]. The higher proportion of healthcare provider's adherence to IPPCG in this study might be due to the time gap. In addition, workplace, and profession-related factors like training, job satisfaction, and relation with the nearby manager and interest to work in the delivery room may contribute to the variation. In this study, about 38.4%, 31%,

**Table 4. Factors associated with immediate postpartum care guidelines among healthcare provider's at hospitals in Gondar province, northwest Ethiopia, 2020/2021 (n = 406).**

| Variables | Category | Adherence to immediate postpartum care | | COR (95% CI) | AOR (95% CI) |
|---|---|---|---|---|---|
| | | Complete | Incomplete | | |
| Intention to stay in their profession | Yes | 130 | 152 | 2.19 (1.38, 3.47) | 1.59 (0.93, 2.71) |
| | No | 34 | 82 | 1 | 1 |
| Marital status | Married | 123 | 119 | 2.43 (1.6, 3.7) | 1.59 (1.15, 3.31)* |
| | Unmarried | 49 | 115 | 1 | 1 |
| Checkup by nearby manager for immediate postpartum cares | Yes | 138 | 149 | 2.32 (1.46, 3.67) | 1.11 (0.61, 2.02) |
| | No | 34 | 85 | 1 | 1 |
| Media exposure | Exposed | 158 | 195 | 2.26 (1.6, 3.7) | 0.86 (0.38, 1.98) |
| | Unexposed | 14 | 39 | | |
| Having smartphone or computer | Yes | 128 | 128 | 2.41 (1.57, 3.69) | 1.49 (0.91, 2.44) |
| | No | 44 | 106 | 1 | 1 |
| Satisfaction on their job | Satisfied | 149 | 175 | 2.84 (1.28, 3.71) | 1.19 (0.63, 2.25) |
| | Unsatisfied | 23 | 59 | 1 | 1 |
| Presence of internet connection in the facility | Yes | 112 | 120 | 1.77 (1.18, 2.66) | 0.79 (0.47, 1.35) |
| | No | 60 | 114 | 1 | 1 |
| Presence of immediate postpartum care guidelines in the ward | Yes | 122 | 96 | 3.51 (2.31, 5.34) | 2.39 (1.44, 3.98)** |
| | No | 50 | 138 | 1 | 1 |
| Presence of assistant for the delivery | Yes | 145 | 173 | 1.89 (1.14, 3.34) | 1.87 (1.10, 3.33)* |
| | No | 27 | 61 | 1 | 1 |
| Received training on essential newborn care | Yes | 84 | 72 | 2.15 (1.43, 3.23) | 0.85 (0.49, 1.46) |
| | No | 88 | 162 | 1 | 1 |
| Received training on BEmONC | Yes | 77 | 49 | 3.06 (1.98, 4.73) | 2.1 (1.2, 3.6)* |
| | No | 95 | 185 | 1 | 1 |
| Age of healthcare providers in year | ≤ 25 | 31 | 54 | 1 | 1 |
| | 26–30 | 102 | 145 | 1.23 (0.74, 2.04) | 0.88 (0.49, 1.57) |
| | ≥31 | 39 | 35 | 1.94 (1.03, 3.66) | 1.66 (0.68, 4.04) |
| Experience in year | ≤2 | 47 | 93 | 1 | 1 |
| | 3–5 | 112 | 98 | 2.26 (1.45, 3.52) | 1.14 (0.66, 1.97) |
| | ≥6 | 13 | 43 | 0.59 (0.29, 1.22) | 0.15 (0.06, 0.38) |
| Average monthly income | <5000 ETB | 45 | 95 | 1 | 1 |
| | 5001–10000 ETB | 114 | 125 | 1.92 (1.24, 2.97) | 1.2 (0.722, 2.02) |
| | ≥10001 ETB | 13 | 14 | 1.96 (0.85, 4.51) | 3.55 (1.30, 9.67)* |

Notes:

* P≤0.05,

**P ≤0.001.

Abbreviations: AOR, adjusted odds ratio; BEmONC, basic emergency obstetric and neonatal care; COR, crude odds ratio; CI, confidence interval; ETB, Ethiopia birr; 1, reference category.

and 71.4% of healthcare providers received training on essential newborn care, BEmONC, and intended to stay in the profession, respectively. Empirical evidence indicated that training regarding maternal and neonatal health is very crucial and help professional to comply with the available standards [25, 28]. In the current study, the healthcare provider's adherence to neonatal length and head circumference measurements were found to be low, which was 48.8% and 49% respectively. This might be due to the perception that length and head circumference are not acute and determinant for the newborn. In this regard, we strongly

recommend healthcare providers to perform all components of immediate postpartum care standards. It then helps to decrease maternal and neonatal mortality and morbidity that occur during the PP.

The present study revealed that the odds of having complete adherence to IPPCG were 2.1 times higher among healthcare providers who have received BEmONC training than healthcare workers who hadn't received BEmONC training. This is due to the fact that BEmONC training is constantly provided to trainees with the aim of fostering basic knowledge and skills in the management of emergency obstetric and neonatal health complications. Therefore, health professionals receiving BEmONC training are likely to have more knowledge and skills about postpartum services, thereby comply with the respective guidelines as compared to non-trained health professionals. In connection to this, existing evidence support that obstetric care providers who received in-service training are more likely to know and practice essential newborn care satisfactorily [29]. Moreover, BEmONC training is essential to combat unacceptably high maternal and neonatal mortalities, particularly in developing countries [28]. This suggests the need for providing BEmONC training for healthcare providers working at maternity and neonatal units regularly, thereby moving forward to achieve the SDGs.

The availability of postpartum care guidelines at the maternity ward is an important determinant factor for healthcare provider's complete adherence to IPPCG. Healthcare providers who had an access to IPPCG at the hospital were 2.39 times more likely to adhere to IPPCG compared with those healthcare providers who haven't accessed the protocol. This might be due to the logic that if instructions are placed in the delivery room in the form of a document or posted on the wall, health professionals can then easily oversee frequently, recall and demonstrate each step proficiently over time. Along with training, IPCG guidelines should be given to health institutions and individual healthcare providers in the form of soft copy or printed material to enhance retention of knowledge and skill.

The odds of having complete adherence to IPPCG among married healthcare providers were 1.59 times higher as compared to those who were not married. This can be possibly explained by married health professionals who will take duties diligently and can act responsibly for everything. Evidence indicated that married healthcare providers have a positive attitude towards reproductive health services compared with unmarried individuals [30]. Likewise, this study found that healthcare providers who had an assistant for the delivery were 1.87 times more likely to have had complete adherence to the immediate postpartum care guidelines as compared to those healthcare providers who hadn't have a birth assistant for the delivery. This is because if there is no assistant during childbirth, the providers may not be able to follow the guidelines properly due to workload or may ignore it altogether. In this regard, allocating sufficient skilled birth attendants at the maternity ward would help provider's adherence to guidelines. Because it curtails task overload and facilitates teamwork, thus early recognition and treatment of complications in case of emergency.

Year of service was found to be significantly associated with healthcare provider's adherence to IPPG. Healthcare providers having 6 and above years of experience had reported 85% likely reduced adherence IPCG than providers with two or less years of experience. We expect that as work experience increases, knowledge, and skills will also increase, and health professionals will be able to fully adhere to IPPCG. However, the finding of this study could be justified, as the work experience increases, the healthcare provider may be looking for more incentives, promotion issues, and educational opportunities. If they do not get the answer they are looking for, they may want to leave the profession, and thereby they may not pay attention to each step of the IPPCG. Empirical evidence supports that the higher the work experience, the more chance of experiencing burn out and intention to leave their profession [31, 32].

Lastly, the current study determined that healthcare providers having a monthly income of greater than 10001 ETB were 3.55 times more likely to perform the postpartum care guidelines properly as compared to healthcare providers who had a monthly income of less than 5000 ETB. This is possibly due to the fact that a balanced salary inspires healthcare provider's not only to love their profession but also encourages them to do all the work properly [33]. Unable to get appropriate payment for their work may cause a hopeless future and overall dissatisfaction to healthcare providers. Hence, health policymakers and other stakeholders need to give special consideration to the benefit packages of healthcare providers at different levels.

## Limitations of the study

The cross-sectional nature of the study design may not possible to generalize the causal relationship between healthcare provider's adherence to IPPG and the hypothesized predictor variables. The study answers reflect on a single moment in time that we could not verify the true characteristics of participants and may not be generalizable over time fluctuations in adherence. Also, this study did not address factors related to the health care system and policies that are designed to plan and provide medical care for people. Besides, in this study, individual and facility level factors were analyzed collectively as an independent variable which may not account for the cluster effect. Moreover, we didn't find enough literature to compare our results with others and make it difficult to discuss in detail. Despite these limitations, the finding of this study provides valuable information regarding healthcare provider's adherence to IPPG.

## Conclusion

The healthcare provider's adherence to IPPCG was found to be low. Having birth assistants, being married, availability of postpartum care guidelines at the maternity ward, received BEmONC training, and higher monthly income was positively associated with healthcare provider's adherence to IPPCG. Whereas, having higher professional working experience was negatively associated with healthcare provider's adherence to IPPCG. Thus, the federal ministry of health and the regional health bureau should pay a special emphasis on hiring adequate health professionals in the maternity ward, constant provision of training, and praiseworthy payments for healthcare providers.

## Supporting information

**S1 Questionnaire. English version of the questionnaire.**
(DOCX)

**S1 Data. SPSS dataset.**
(SAV)

**S1 File. Observation guide for data collectors.**
(DOCX)

## Acknowledgments

We would like to thank the University of Gondar for providing study ethical clearance to conduct this study. Our gratitude also goes to all data collectors and study participants. We are glad to Hospitals in Gondar province for writing permission letter.

## Author Contributions

**Conceptualization:** Azmeraw Ambachew Kebede, Getachew Azeze Eriku.

**Data curation:** Azmeraw Ambachew Kebede, Birhan Tsegaw Taye, Kindu Yinges Wondie, Agumas Eskezia Tiguh, Getachew Azeze Eriku, Muhabaw Shumye Mihret.

**Formal analysis:** Azmeraw Ambachew Kebede, Birhan Tsegaw Taye, Kindu Yinges Wondie, Agumas Eskezia Tiguh, Getachew Azeze Eriku, Muhabaw Shumye Mihret.

**Funding acquisition:** Azmeraw Ambachew Kebede, Birhan Tsegaw Taye, Kindu Yinges Wondie, Agumas Eskezia Tiguh, Getachew Azeze Eriku, Muhabaw Shumye Mihret.

**Investigation:** Azmeraw Ambachew Kebede, Birhan Tsegaw Taye, Getachew Azeze Eriku, Muhabaw Shumye Mihret.

**Methodology:** Azmeraw Ambachew Kebede, Birhan Tsegaw Taye, Kindu Yinges Wondie, Agumas Eskezia Tiguh, Getachew Azeze Eriku, Muhabaw Shumye Mihret.

**Project administration:** Azmeraw Ambachew Kebede, Birhan Tsegaw Taye, Kindu Yinges Wondie, Agumas Eskezia Tiguh, Getachew Azeze Eriku, Muhabaw Shumye Mihret.

**Resources:** Azmeraw Ambachew Kebede, Birhan Tsegaw Taye, Kindu Yinges Wondie, Agumas Eskezia Tiguh, Getachew Azeze Eriku, Muhabaw Shumye Mihret.

**Software:** Azmeraw Ambachew Kebede, Birhan Tsegaw Taye, Kindu Yinges Wondie, Agumas Eskezia Tiguh, Getachew Azeze Eriku, Muhabaw Shumye Mihret.

**Supervision:** Azmeraw Ambachew Kebede, Birhan Tsegaw Taye, Kindu Yinges Wondie, Agumas Eskezia Tiguh, Getachew Azeze Eriku, Muhabaw Shumye Mihret.

**Validation:** Azmeraw Ambachew Kebede, Birhan Tsegaw Taye, Kindu Yinges Wondie, Agumas Eskezia Tiguh, Getachew Azeze Eriku, Muhabaw Shumye Mihret.

**Visualization:** Azmeraw Ambachew Kebede, Birhan Tsegaw Taye, Kindu Yinges Wondie, Agumas Eskezia Tiguh, Getachew Azeze Eriku, Muhabaw Shumye Mihret.

**Writing – original draft:** Azmeraw Ambachew Kebede.

**Writing – review & editing:** Azmeraw Ambachew Kebede, Birhan Tsegaw Taye, Kindu Yinges Wondie, Agumas Eskezia Tiguh, Getachew Azeze Eriku, Muhabaw Shumye Mihret.

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
