## [Decision Letter · Decision Letter 0]

28 Jul 2021

PONE-D-21-13480

Healthcare provider’s adherence to immediate postpartum care guidelines in Gondar province hospitals, northwest Ethiopia: A multicenter study

PLOS ONE

Dear Dr. Kebede,

Thank you for submitting your manuscript to PLOS ONE. After careful consideration, we feel that it has merit but does not fully meet PLOS ONE’s publication criteria as it currently stands. Therefore, we invite you to submit a revised version of the manuscript that addresses the points raised during the review process.

We look forward to receiving your revised manuscript.

Kind regards,

Frank T. Spradley

Academic Editor

PLOS ONE

Journal Requirements:

Reviewers' comments:

Reviewer's Responses to Questions

**Comments to the Author**

1. Is the manuscript technically sound, and do the data support the conclusions?

Reviewer #1: No

Reviewer #2: Yes

2. Has the statistical analysis been performed appropriately and rigorously? 

Reviewer #1: No

Reviewer #2: Yes

3. Have the authors made all data underlying the findings in their manuscript fully available?

Reviewer #1: No

Reviewer #2: Yes

4. Is the manuscript presented in an intelligible fashion and written in standard English?

Reviewer #1: No

Reviewer #2: Yes

5. Review Comments to the Author

Reviewer #1: Thank you for providing this opportunity to review the paper. My comments are listed below, according to the displayed order of the manuscripit.

INTRODUCTION

The main sentence is not well corresponded with the sentence in the parenthesis.

> Sub-Saharan Africa (SSA) merely accounted for more than two-thirds of the global MM (i.e., 542 maternal deaths per 100, 000 live births)

Please define what PPP means.

> Thus, about 20 to 44% of these maternal deaths occur during the PPP [8].

We need relevant information to understand the fact of 'low healthcare access'.

> Ethiopia is one of the SSA countries with the highest maternal and neonatal mortalities and low healthcare access.

Several words - professional, providers, midwife - are repeatedly appeared in this sentence. Pleare require reconsiderations.

> To reduce maternal and neonatal mortalities on a permanent and sustainable basis, healthcare professionals, especially midwives, play the lion’s share [18] as healthcare providers, in specific midwives, are the frontline care providers for women and newborns [19].

METHODS

This sentence has several verbs, so please correct it.

> These include medical doctors, midwives, and integrated emergency surgeon officers (IESO) working at hospitals in northwest Ethiopia during the data collection period were included.

In the sample size calculation, how do the authors consider cluster effect? Because each facility can be considered as a cluster which affected to practices of the providers in the same facility as the authors discussed in this manuscript.

Definition of several variables and data processing procedure are not clear. For instance, how the satisfaction was treated only into two categories despite of several questions; is dichotomization of the answer to 'self-rated relation with the nearby boss' sufficent?; criteria to assess 'proper documation' is not clear; what does 'Presence of assistant for the delivery' mean, who was the assistant, what was the standard role of the assistant; oxytocin provided to women - when and how?; APGAR score at when?; Uterine massage - when and how? etc. Please add sufficient explanation in the main text, or in annex.

How the observation was conducted should be described. For instance, just one case of care by a provider was observed, or several observations were used for the assessment.

I do not understand what '5%' means.

> Before the actual data collection, a pretest was done on 5% of healthcare providers outside of the study area.

RESULTS

In the first part of the results section, the number of recruited participants, the number of actual participants, and respondent rate should be described.

This sentence is not complete. Does it show the proportion of providers who performed full adherence?

> The proportion of healthcare provides adhering to IPPCG was 42.4% (95% CI: 37.5, 47.2).

I recommend the authors to modify Table 3. Please display the items according to the time from the birth up to six hours of postpartum.

Was there any consideration of collineality in the multiple logistic regression model? If yes, how was it evaluated and treated? I guess that age and experience should have a positive linear relationship. Media exposure and having PC or mobile is as well.

DISCUSSION

I am not convinced by this speculation:

> The discrepancy might be due to variation in the study population. The study from Mekelle includes nurses and midwives whereas our study incorporates midwives, medical doctors, and IESO.

Because the proportion of medical doctors and IESO in this study is less than 5%. The authors did not show relationship between the qualification and the adherence, so the claim is not valid by the data.

Reviewer #2: In my opinion the article is very well written and the survey was planned well. Authors clearly indicates the scope of the rules to be followed when caring for the mother and the newborn baby immediately after childbirth. The authors paid attention to the recommendations in the field of prophylaxis and their implementation.

6. PLOS authors have the option to publish the peer review history of their article (what does this mean?). If published, this will include your full peer review and any attached files.

Reviewer #1: No

Reviewer #2: No

---

## [Decision Letter · Decision Letter 1]

1 Sep 2021

PONE-D-21-13480R1

Healthcare provider’s adherence to immediate postpartum care guidelines in Gondar province hospitals, northwest Ethiopia: A multicenter study

PLOS ONE

Dear Dr. Kebede,

Thank you for submitting your manuscript to PLOS ONE. After careful consideration, we feel that it has merit but does not fully meet PLOS ONE’s publication criteria as it currently stands. Therefore, we invite you to submit a revised version of the manuscript that addresses the points raised during the review process.

A rebuttal letter that responds to each point raised by the academic editor and reviewer(s). You should upload this letter as a separate file labeled 'Response to Reviewers'.A marked-up copy of your manuscript that highlights changes made to the original version. You should upload this as a separate file labeled 'Revised Manuscript with Track Changes'.An unmarked version of your revised paper without tracked changes. You should upload this as a separate file labeled 'Manuscript'

We look forward to receiving your revised manuscript.

Kind regards,

Frank T. Spradley

Academic Editor

PLOS ONE

Reviewers' comments:

Reviewer's Responses to Questions

**Comments to the Author**

1. If the authors have adequately addressed your comments raised in a previous round of review and you feel that this manuscript is now acceptable for publication, you may indicate that here to bypass the “Comments to the Author” section, enter your conflict of interest statement in the “Confidential to Editor” section, and submit your "Accept" recommendation.

Reviewer #1: (No Response)

2. Is the manuscript technically sound, and do the data support the conclusions?

Reviewer #1: Partly

3. Has the statistical analysis been performed appropriately and rigorously? 

Reviewer #1: No

4. Have the authors made all data underlying the findings in their manuscript fully available?

Reviewer #1: Yes

5. Is the manuscript presented in an intelligible fashion and written in standard English?

Reviewer #1: Yes

6. Review Comments to the Author

Reviewer #1: I do not understand why the authors stated 'the effect of inter-facility practice will not be problem' in the response without showing any evidence. This is not right attitude as a peer-review process in a scientific paper.

If we see the positive result in this article, it is apparent that setting of each facility affected to the outcome, since 'availability of postpartum care guidelines at the maternity ward' has significant effect. The authors mixed up individual factors (i.e. marital status, income) and facility factors (i.e. presence of internet connection, presence of guidelines) in its analysis. So that individual response cannot be treated individually. In addition, as I have pointed out, the reasons why cluster effect was not considered in the study design should be clearly explained.

Regarding the job satisfaction, the referred paper (Lu Y et al.) used eight questions while this article mentioned that there were nine questions. An explanation on this difference is expected.

The authors mentioned that 'one provider was observed only once'. Discussion on its validity considering possible fluctuations in individual practices is expected.

Following sentence provides a confusion on mortality, mortality rate, and mortality ratio, since MM and NM were not indicated as ratio nor rate. Please revise them:

Thus, the MM was 871 in 2000 [13], 673 in 2005 [14], 676 in 2011 (15), and 412 per 100,000 live births in 2016 [12]. Besides, neonatal mortality (NM) has dropped from 49 in 2000 to 39 in 2005 [13,14], and 37 in 2011 to 29 per 1000 live births in 2016 [12,15]. In Ethiopia, as a country, a lot of works has been done to increase maternal healthcare service utilization with the promise of reducing the maternal mortality ratio from 412 per 100,000 live births to 199, and the neonatal mortality rate from 29 to 10 per 1000 live births by 2020 [16].

Careless mistakes are found. Significant digits in each table and main texts should be unified. Odds ratio and its CI 1.19 (06.3, 2.25) is an apparent mistake, 0.15 (0.06, 0.38) is not significant? Reference list should be reconsidered to follow a standard of the journal.

7. PLOS authors have the option to publish the peer review history of their article (what does this mean?). If published, this will include your full peer review and any attached files.

Reviewer #1: No

---

## [Decision Letter · Decision Letter 2]

4 Oct 2021

PONE-D-21-13480R2Healthcare provider’s adherence to immediate postpartum care guidelines in Gondar province hospitals, northwest Ethiopia: A multicenter studyPLOS ONE

Dear Dr. Kebede,

Thank you for submitting your manuscript to PLOS ONE. After careful consideration, we feel that it has merit but does not fully meet PLOS ONE’s publication criteria as it currently stands. Therefore, we invite you to submit a revised version of the manuscript that addresses the points raised during the review process.

Please respond to the reviewer's minor comments. Also, correct the statement that you emailed me about earlier.

Submit your revised manuscript by Nov 18 2021 11:59PM. If you will need more time than this to complete your revisions, please reply to this message or contact the journal office at plosone@plos.org. Please include the following items when submitting your revised manuscript:A rebuttal letter that responds to each point raised by the academic editor and reviewer(s). You should upload this letter as a separate file labeled 'Response to Reviewers'.A marked-up copy of your manuscript that highlights changes made to the original version. You should upload this as a separate file labeled 'Revised Manuscript with Track Changes'.An unmarked version of your revised paper without tracked changes. You should upload this as a separate file labeled 'Manuscript'.

We look forward to receiving your revised manuscript.

Kind regards,

Frank T. Spradley

Academic Editor

PLOS ONE

Journal Requirements:

Reviewers' comments:

Reviewer's Responses to Questions

**Comments to the Author**

1. If the authors have adequately addressed your comments raised in a previous round of review and you feel that this manuscript is now acceptable for publication, you may indicate that here to bypass the “Comments to the Author” section, enter your conflict of interest statement in the “Confidential to Editor” section, and submit your "Accept" recommendation.

Reviewer #1: (No Response)

2. Is the manuscript technically sound, and do the data support the conclusions?

Reviewer #1: Partly

3. Has the statistical analysis been performed appropriately and rigorously? 

Reviewer #1: No

4. Have the authors made all data underlying the findings in their manuscript fully available?

Reviewer #1: Yes

5. Is the manuscript presented in an intelligible fashion and written in standard English?

Reviewer #1: Yes

6. Review Comments to the Author

Reviewer #1: Dear authors,

Thank you for responding to previous comments. Almost all the concerns have been solved except one. I would appreciate it if the authors would state counter-arguments on the following issue.

The authors explained why they have to keep information on hospital names confidential in several sentences. But this does not make a sense. Because my previous comment was why the authors do not take cluster effect into account in the analysis. This can be performed without indicating each hospital and participant.

The authors explained that 'the entire population of the study are internally homogenous (i.e. health care provider’s level of qualification was almost the same)'; however, this statement is still arbitrary, since no data shows the homogeneity of the participants.

In addition, as I have pointed out, the analysis mixed individual and facility factors as the independent variables. One possible solution would be using 'multilevel logistic regression model' instead of binary one.

7. PLOS authors have the option to publish the peer review history of their article (what does this mean?). If published, this will include your full peer review and any attached files.

Reviewer #1: No

---

## [Decision Letter · Decision Letter 3]

18 Oct 2021

Healthcare provider’s adherence to immediate postpartum care guidelines in Gondar province hospitals, northwest Ethiopia: A multicenter study

PONE-D-21-13480R3

Dear Dr. Kebede,

We’re pleased to inform you that your manuscript has been judged scientifically suitable for publication and will be formally accepted for publication once it meets all outstanding technical requirements.

Kind regards,

Frank T. Spradley

Academic Editor

PLOS ONE

Reviewers' comments:

Reviewer's Responses to Questions

**Comments to the Author**

1. If the authors have adequately addressed your comments raised in a previous round of review and you feel that this manuscript is now acceptable for publication, you may indicate that here to bypass the “Comments to the Author” section, enter your conflict of interest statement in the “Confidential to Editor” section, and submit your "Accept" recommendation.

Reviewer #1: All comments have been addressed

2. Is the manuscript technically sound, and do the data support the conclusions?

Reviewer #1: Yes

3. Has the statistical analysis been performed appropriately and rigorously? 

Reviewer #1: Yes

4. Have the authors made all data underlying the findings in their manuscript fully available?

Reviewer #1: Yes

5. Is the manuscript presented in an intelligible fashion and written in standard English?

Reviewer #1: Yes

6. Review Comments to the Author

Reviewer #1: (No Response)

7. PLOS authors have the option to publish the peer review history of their article (what does this mean?). If published, this will include your full peer review and any attached files.

Reviewer #1: No

---

## [Editor Report · Acceptance letter]

20 Oct 2021

PONE-D-21-13480R3 

Healthcare provider’s adherence to immediate postpartum care guidelines in Gondar province hospitals, northwest Ethiopia: A multicenter study 

Dear Dr. Kebede:

I'm pleased to inform you that your manuscript has been deemed suitable for publication in PLOS ONE. Congratulations! Your manuscript is now with our production department. 

Kind regards, 

on behalf of

Dr. Frank T. Spradley 

Academic Editor

PLOS ONE